# Modeling the potential effects of rooftop solar on household energy burden in the United States

Sydney P. Forrester [1] ✉, Cristina Crespo Montañés[1], Eric O'Shaughnessy[1] & Galen Barbose [1]

Policymakers at the federal and state level have begun to incorporate energy burden into equity goals and program evaluations, aiming to reduce energy burden below a high level of 6% for lower income households in the United States. Pairing an empirical household-level dataset spanning United States geographies together with modeled hourly energy demand curves, we show that rooftop solar reduces energy burden across a majority of adopters during our study period from a median of 3.3% to 2.6%. For low- and moderate-income adopters (at or below 80% and 120% of area median income, respectively), solar reduces median 2021 energy burden from 7.7% to 6.2%, and 4.1% to 3.3%, respectively. Importantly, solar reduces the rate of high or severe energy burden from 67% of all low-income households before adoption to 52% of households following adoption, and correspondingly from 21% to 13% for moderate-income households. Here, we show rooftop solar can support policy goals to reduce energy burden along with strategies such as weatherization and bill assistance.

Spatial and structural inequities in our energy system have led some communities to benefit while others bear the burden of its byproducts, such as local air pollution[1] or cost shifts[2]. In particular, energy affordability and access to supporting technologies, such as energy efficiency retrofits or rooftop solar photovoltaics (PV), are distributed unevenly across United States (U.S.) households[3–10]. To target assistance to those households most vulnerable to increasing energy prices, policymakers are introducing indicators to quantify and track energy equity[11]. One common energy affordability metric is energy burden (EB), or the percentage of gross income that a household spends on energy costs[10,12–14]. Typically, an EB above 6% is considered high, while an EB above 10% is considered severe[3].

Existing EB literature has focused on quantification across populations. For example, Drehobl et al. found that low-income households experienced an EB triple the magnitude of non-low- to moderate-income (LMI) households[3]. Studies have correlated higher EB with fuel oil heating[6], specific climate zones[6], minority households[3,4,7,15], households with concurrent high housing burdens[7], and island or rural

populations[5]. A high EB has been linked to health impacts as well[8], such as a higher level of winter deaths in cold climate regions[16] and other conditions associated with poor housing stock such as respiratory illness, thermal discomfort, and mental health stressors associated with the ability to pay[14]. EB is becoming more common in U.S. policy at the state and federal levels. For example, the Department of Energy identified eight policy priorities, with the first to decrease the energy burden in disadvantaged communities, as a result of the Justice40 initiative[17,18]. Additionally, several states are shifting energy affordability efforts and energy efficiency policies to focus on EB reduction for lower-income residents[11,13,19].

Several tools exist to increase energy affordability. Historically, most have focused on either providing short-term bill assistance and discounts or longer-term energy efficiency home retrofits. Energy insecurity has long been linked to low levels of home energy efficiency in the body of literature[5,14,20–22]. In addition, weatherization can create sustained bill savings as well as generate non-energy benefits[16,22,23]. As a result, there is a strong body of weatherization research and policy to

[1]Lawrence Berkeley National Laboratory, 1 Cyclotron Road, Berkeley, CA 94720, USA. ✉e-mail: SPForrester@lbl.gov

reduce EB[4,6,24]. More recently, work focusing on the potential for rooftop solar adoption to reduce EB has emerged as another option for sustained bill reduction as solar adoption by LMI households has increased, in part due to decreasing costs and LMI solar programs[25–28]. Solar, weatherization, and other methods of sustained net energy reduction are important since they reduce household exposure to potential increases in energy prices[29].

At present, few studies quantify the impact of rooftop solar on EB. Those that do, focus on a specific location and/or use aggregated data to make assumptions about income and energy costs[9,10,30,31]. Tidemann et al. found that the use of aggregated data to explore the distributional outcomes of rooftop solar led to results that overlooked vulnerable areas[32]. Indeed, the lack of micro-level data can be a large challenge[33,34]. This paper fills several important existing gaps in the literature. We explore how residential rooftop solar impacts EB (electric and non-electric) at the household level across the U.S. This paper also leverages empirical data wherever possible. In contrast, other studies have relied on simplifications such as Kerby et al. (2024), which assigns uniform solar system sizes and tariffs and does not include energy burden results due to a lack of income data[31]. Furthermore, some existing literature on EB reduction from solar focuses solely on the bill impacts without accounting for ongoing off-bill financial impacts of solar adoption, such as ongoing solar loan or lease payments or ongoing production-based incentive payments based on solar generation[30]. Focusing exclusively on bill impacts of rooftop PV overly simplifies the economic impacts of solar adoption for homes and can lead to overestimates of EB reductions.

In this work, we consider a broader scope of financial impacts when estimating EB during our study period of 2021, including both the direct on-bill impacts as well as off-bill impacts, to show that rooftop solar reduced energy burden for the majority of adopters, including low- and moderate-income adopters here defined as adopters at or below 80% area median income (AMI) and 120% AMI, respectively. We henceforth use EB to characterize the percentage of household income spent on energy costs, inclusive of costs and incentives (unless otherwise specified).

## Results
Energy burden impacts of rooftop solar were analyzed across ownership models, income groups, year of solar adoption, region, and heating fuel type. Figure 1 shows the average change in EB across each group's population with respective 95% confidence intervals. This illustrates the differential impact of these factors on EB reduction. We provide further detail on these findings in the subsections below and include results from analysis of variance (ANOVA), t-tests, and Monte Carlo analyses throughout this section as well as in Supplementary Note 2 and Supplementary Fig. 5 that demonstrate the robustness of our results to variation and that each group is statistically different from one another.

### Energy burden impacts in 2021 for the study population
After rooftop solar installation, energy bills for the entire sample of adopters shifted from a median of 3.3% to 1.3% of gross income. However, taking off-bill loan and lease repayment into account muted the downward impact of rooftop solar on a household's EB, while ongoing benefits such as renewable energy certificates for host-owned systems amplified savings. Altogether, adopters in our sample settled to a median EB of 2.6%. From the perspective of customers, the median customer saw a 1.7 EB point reduction when looking at bill savings alone ($1987 annually) versus a 0.6 point reduction ($691 annually) when considering off-bill financial impacts. When off-bill impacts are excluded, the effect of solar installation on burden appears to be over three times as large as it actually is, illustrating the overestimation of rooftop solar EB reduction potential that may occur if off-bill impacts are disregarded.

There were a number of households whose off-bill costs exceeded their bill savings in the analysis period of 2021, and thus, their EB increased as a result of solar (Fig. 2). There are several reasons why this may be the case. First, since this was only a snapshot of 2021 impacts, it does not inform whether a system is economical over the course of its lifespan, and an adopter may initially pay a premium to hedge against future rate increases. Indeed, Supplementary Fig. 2 shows a positive net present value when looking solely at 2021 adopters, regardless of cash, loan, or lease payment. Second, a number of our financial assumptions related to the loan and lease terms are based on state averages at the year of adoption and may not reflect an adopter's precise terms. Third, an adopter may also choose solar for other reasons apart from economics.

We explore this topic further in Supplementary Note 1. Supplementary Table 10 shows that a low EB absent solar, low electricity prices, larger solar system sizes, more expensive per-watt solar costs, smaller home square footage, and higher income all were linked to lower EB reduction (in that order of magnitude). We also find that homes using fuel oil or propane as a primary heating fuel were more likely to see low levels of savings, as were homes located in the South.

### Energy burden impacts in 2021 by ownership model
Each customer was assumed to finance its system through either a loan or lease. The 56% of customers with host-owned (loan financed) systems saw a median burden of 3.0%, absent solar, decrease to 2.4%, with individuals seeing a median 0.50 EB point drop ($660 annually). Those with third-party owned (leased) systems saw median burdens fall from 3.7% to 3.0%, with individuals on average seeing a 0.63 point reduction ($716 annually) (Fig. 3). On average, solar adoption reduced household energy burdens for systems financed through lease payments by roughly 0.2 percentage points more than for systems financed through loans ($t = -38.3$). Though customers who leased solar panels saw larger point reductions in EB, they also have relatively higher energy burdens absent solar, which may be linked to the fact that a higher share of third-party-owned systems is correlated with lower incomes[25].

Detailed results from a sensitivity analysis are depicted in Supplementary Fig. 4 and suggest that the results are generally more sensitive to our leasing assumptions. On the other hand, varying loan assumptions have relatively little impact on the results.

### Energy burden impacts in 2021 by income group
Across the 500k households in the study sample, 23% were low-income (≤80% AMI), 21% were moderate-income (80–120% AMI), and 57% were non-LMI (>120% AMI). Without solar, 34% of these low-income households would have experienced a severe EB of over 10%, while an additional 32% would have experienced a high EB of 6–10%. By comparison, the corresponding numbers for moderate-income households experienced were 2.6% and 19%. and those for non-LMI households were 0.25% and 2.7% (Fig. 4). With solar, the number of low-income households experiencing severe and high EB experienced a percentage point drop of 8.8 and 5.5, respectively, while moderate-income households saw corresponding point drops of 0.4 and 8.5, respectively. Overall, solar was able to reduce EB to manageable levels below 6% for 36% of the subset of LMI adopters experiencing severe or high burden.

Solar adoption reduced low-income household energy burden by roughly 1.3 percentage points more than for high-income households ($F = 15061.9$, $p < 0.0005$). More specifically, median EB decreased from 7.7% to 6.2% for low-income adopters and from 4.1% to 3.3% for moderate-income adopters (Fig. 4). EB above 6% with solar may be linked with ongoing difficulty to pay for energy, indicating that solar alone may not be able to fully alleviate high EB for the majority of low-income adopters and a smaller group of moderate-income adopters. Consequently, households with post-adoption burdens exceeding 6% may require additional measures (e.g., energy efficiency) or incentives

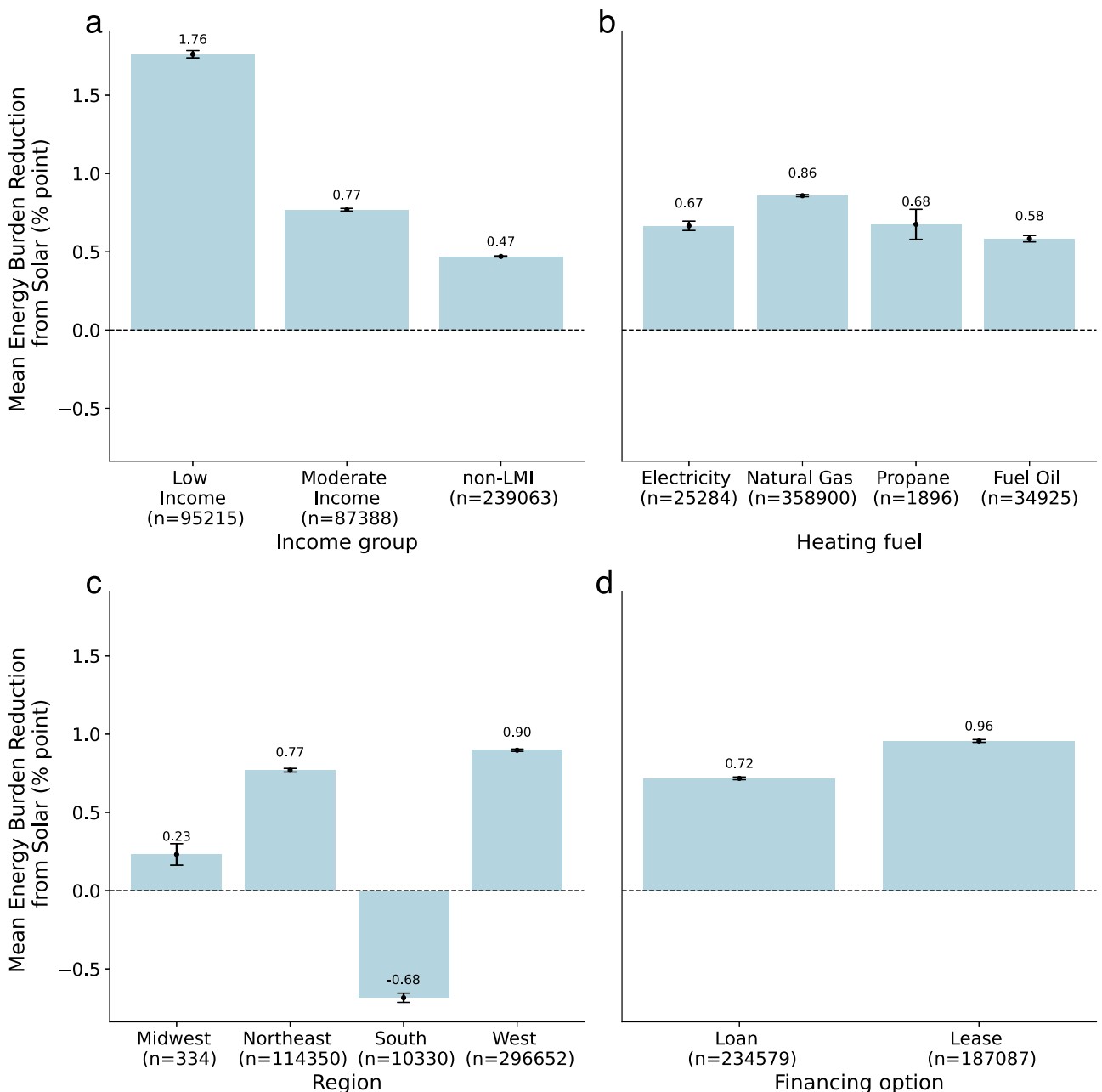

**Fig. 1 | Mean energy burden percentage point reduction due to rooftop solar adoption.** Shown for households across **a** income, **b** primary heating fuel type, **c** region, and **d** financing structure. Data are presented as mean values ± SEM. Source data are provided with this paper.

(e.g., low-income solar programs) to reduce all LMI households' EB below 6%. Figure 4 further illustrates the importance of incorporating off-bill impacts into EB, particularly for lower-income groups. For example, rooftop solar reduced the share of LMI households experiencing high or severe EB from 45% to 33%, however, solely using bill impacts would erroneously imply a reduction to just 15% of low-income households.

### Energy burden impacts in 2021 by year of adoption

This analysis looks at EB impacts for the year 2021 but incorporates empirical cost and incentive information specific to the year of adoption, as well as loan and lease terms from each respective year/state combination. Over time, EB absent solar has increased slightly across the entire sample from 2.9% in 2013 to 3.2% in 2021, consistent with the fact that the incomes of adopters have been slowly migrating downwards[25]. While the 2021 income estimates for low-income 2021

adopters are 6% less than those of 2013 adopters, EB absent solar is 8% higher. Thus, while solar has been increasingly reaching lower-income households, it has been comparatively less successful at increasing adoption among low-income households with higher EB (Fig. 5). This may be due to the broadening of the solar market into lower-income regions, among other factors[25].

### Energy burden impacts in 2021 across regions

Across our sample, the impacts of solar adoption on household EB vary by region. Location-specific impacts include differences in income, solar incentives, cost, and resource, and energy costs impacted by prevalent heating fuel types and energy prices (Table 1). Supplementary Table 10 and Supplementary Note 1 find that controlling for many of these variables still shows regional differences, which could be due to additional factors such as heating degree days, cooling degree days, housing stock, or energy usage behavior.

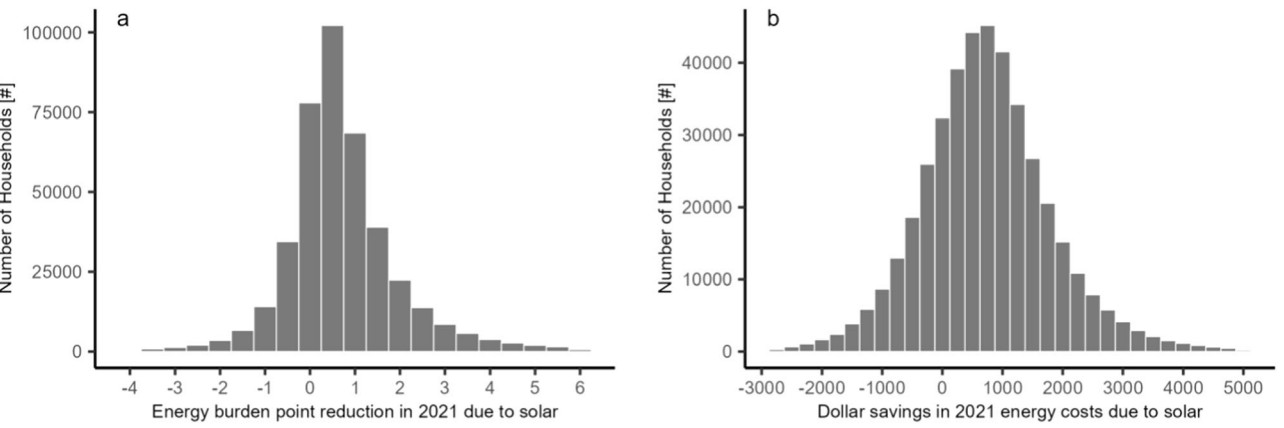

**Fig. 2 | Energy burden (EB) point reduction and dollar savings due to solar (N = 421,666). a** Shows the EB point change across all adopters due to solar while **b** shows the monetary change in energy costs. Both are for the analysis period of 2021, and in both cases, positive values indicate net savings while negative values indicate net costs.

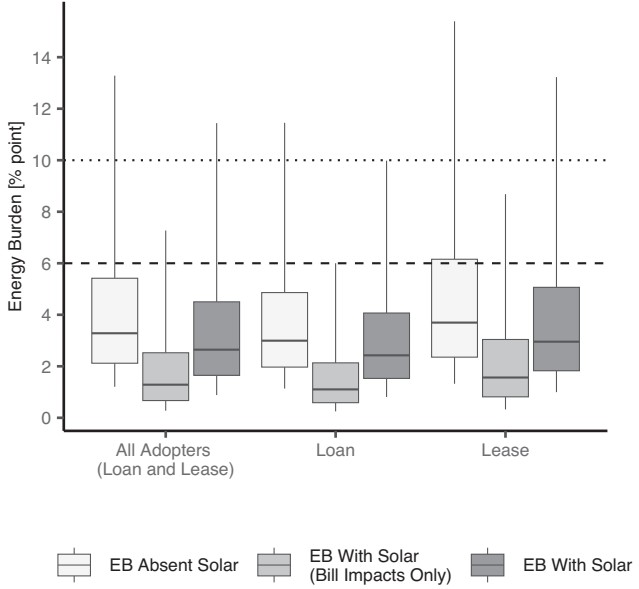

**Fig. 3 | Distribution of energy burden (EB) by ownership structure.** EB distributions across adopter households absent solar (all adopters, N = 421,666) versus with solar either loan-financed (host-owned system adopters, N = 234,579) or leased (third-party owned system adopters, N = 187,087). Mid-line indicates median, lower (upper) box boundaries denote the 25th (75th) percentile, and lower (upper) whiskers are the 5th (95th) percentile. Source data are provided with this paper.

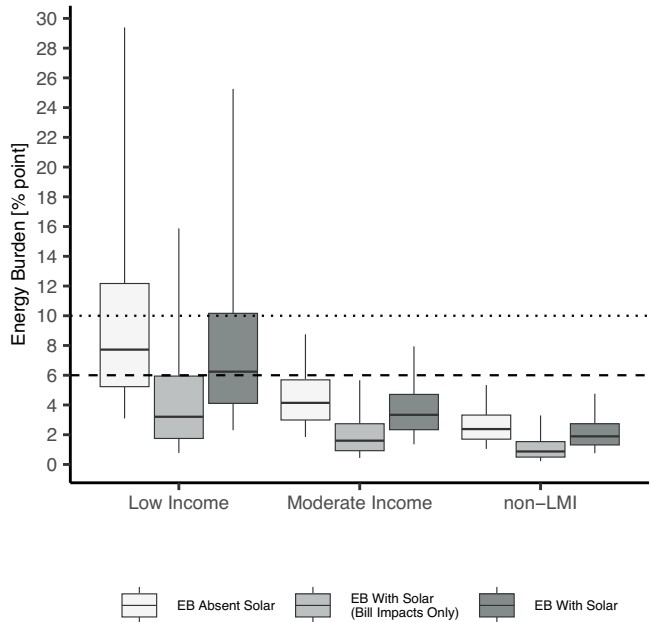

**Fig. 4 | Distribution of energy burden (EB) across low-income (N = 95,215), moderate-income (N = 87,388), and non-LMI (N = 239,063) adopter households absent solar, bill impacts only, and with solar (loan financed or leased).** Low-income groups consist of adopters with household incomes at or below 80% of area median income (AMI); moderate-income adopters are at 80–120% AMI; and non-LMI adopters are over 120% AMI. Mid-line indicates the median, lower (upper) box boundaries denote the 25th (75th) percentile, and lower (upper) whiskers are the 5th (95th) percentile. Dotted line at 6% reflects the cutoff for a high EB, and at 10% reflects the cutoff for a severe EB. Source data are provided with this paper.

Figure 6 shows that rooftop solar, on average, decreases EB across income groups in the Midwest, Northeast, and West. Notably, for low-income adopters, solar was able to push the majority of adopters in the West from a high energy burden (7.3% median) to 5.7% (below the 6% level). However, across all income groups, median EB increases in the South. In all, solar reduced EB for households in the West by roughly 1.6 percentage points more than for households in the South ($F = 2132.1$, $p < 0.0005$). Nevertheless, the energy burden across solar adopter income groups in the South remains comparable to levels in the Midwest and falls below those in the Northeast. The increase in costs in the South is consistent with a study based in Florida[35] and is explored further in Supplementary Note 1. We hypothesize that this is largely due to the relatively low cost of electricity which attenuates the bill reduction potential of rooftop solar. In the Northeast, EB demonstrated higher variation, likely due to the prevalence of propane and fuel oil heating (Table 1), which tend

to be costlier than electricity and natural gas and cannot be offset by solar.

## Energy burden impacts in 2021 by primary heating fuel type

Solar alone can only impact the volumetric cost of electricity. As such, in utility territories with high fixed costs or minimum bills, solar will not fully offset electricity bills. Additionally, rooftop solar alone cannot offset energy costs associated with non-electric loads such as natural gas, propane, or fuel oil heating. Solar adoption reduced EB for households using natural gas as a primary heating fuel by roughly 0.3 percentage points more than those using fuel oil ($F = 157.2$, $p < 0.0005$). Figure 7 illustrates that the highest EB across income groups is for households with either propane or fuel oil heating.

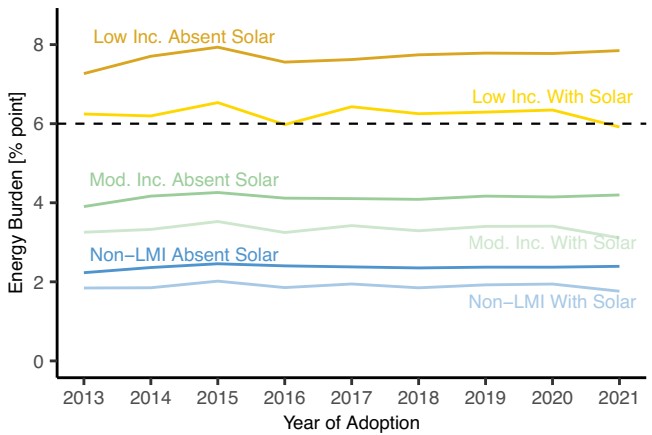

**Fig. 5 | Median energy burden trend line with and without solar, by income group.** 2021 energy burden (EB) distributions by year of adoption before and after solar adoption, by low income ($N = 95,215$), moderate-income ($N = 87,388$), and non-low- or moderate-income (non-LMI) ($N = 239,063$) levels. Dotted line at 6% reflects the cutoff for a high EB. Source data are provided with this paper.

Moreover, almost all households in the small group of non-LMI solar households whose EB remains above 6% have propane or fuel oil heating. Additionally, of all low-income adopters with propane and fuel oil heating, 85% and 80%, respectively, maintain an EB >6%.

Since rooftop solar adoption can only reduce EB a limited amount for those with non-electric heating, strategies such as weatherization may be necessary in order to reduce heating energy requirements regardless of fuel type. Alternatively, the electrification of large non-electric loads such as heating would allow for solar to offset those costs, albeit with a larger system. This is an area for further exploration.

## Discussion

Rooftop solar effectively reduced energy costs for the great majority of U.S. adopters. Across our sample, solar reduced EB from 3.3% to 2.6%. Taking costs into account, as well as revenue from bill savings and incentives when calculating EB, is critical as it avoids over-estimating EB reduction from solar. In contrast, only taking bill savings into account erroneously shows median EB falling to 1.2%.

EB reduction from solar generally persists regardless of the year of adoption, income, region, and fuel type. The one exception is in the South where electricity costs are low, reducing the bill reduction potential of solar and leading to higher costs than revenue. In these cases, solar increases EB, but levels remain low due to relatively inexpensive energy costs. On the other hand, in cases of households with low incomes or heating from propane or fuel oil, solar reduces EB but levels remain high.

From our sample, 67% of low-income households and 21% of moderate-income households experienced either high or severe EB. Solar adoption was able to successfully reduce EB to manageable levels below 6% for a third of households in this group (down to 52% and 13% of households, respectively). Overall, solar decreased median EB for low-income and moderate-income adopters from 7.7% to 6.2% and from 4.1% to 3.3%, respectively.

Interpretation of our results should take into account certain methodological limitations in our research design. First, given data limitations on spatially and temporally granular energy demand profiles, we assign modeled hourly demand to our empirical household and solar system data. While this analysis was only possible with the use of synthetic load data, this can propagate any biases in the underlying physical modeling and forces us to assign loads independently of household income and retail energy prices, effectively supposing income and price elasticities of energy demand equal to one.

**Table 1 | Descriptive statistics of solar adopters by region**

| | Med. area med. income [%] | Med. 2021 Inc. [$1000] | Med. solar size [kW] | Med. solar cost [$/W] | Third-Party owner-ship [%] | Fuel oil or pro-pane heat [%] | Med. electricity price [$/kWh] | Med. 2021 energy costs without solar [$] | Med. energy burden without solar [%] | Med. energy bur-den with solar [%] |
|---|---|---|---|---|---|---|---|---|---|---|
| Midwest | 136 | 120 | 6.6 | 5.05 | 0 | 1 | $0.13 | 3250 | 2.8 | 2.6 |
| Northeast | 123 | 117 | 7.2 | 4.29 | 64 | 30 | $0.20 | 4508 | 4.1 | 3.4 |
| South | 146 | 98 | 6.9 | 4.41 | 0 | 0 | $0.08 | 2138 | 2.1 | 2.7 |
| West | 137 | 121 | 5.8 | 4.64 | 38 | 1 | $0.28 | 3598 | 3.1 | 2.4 |

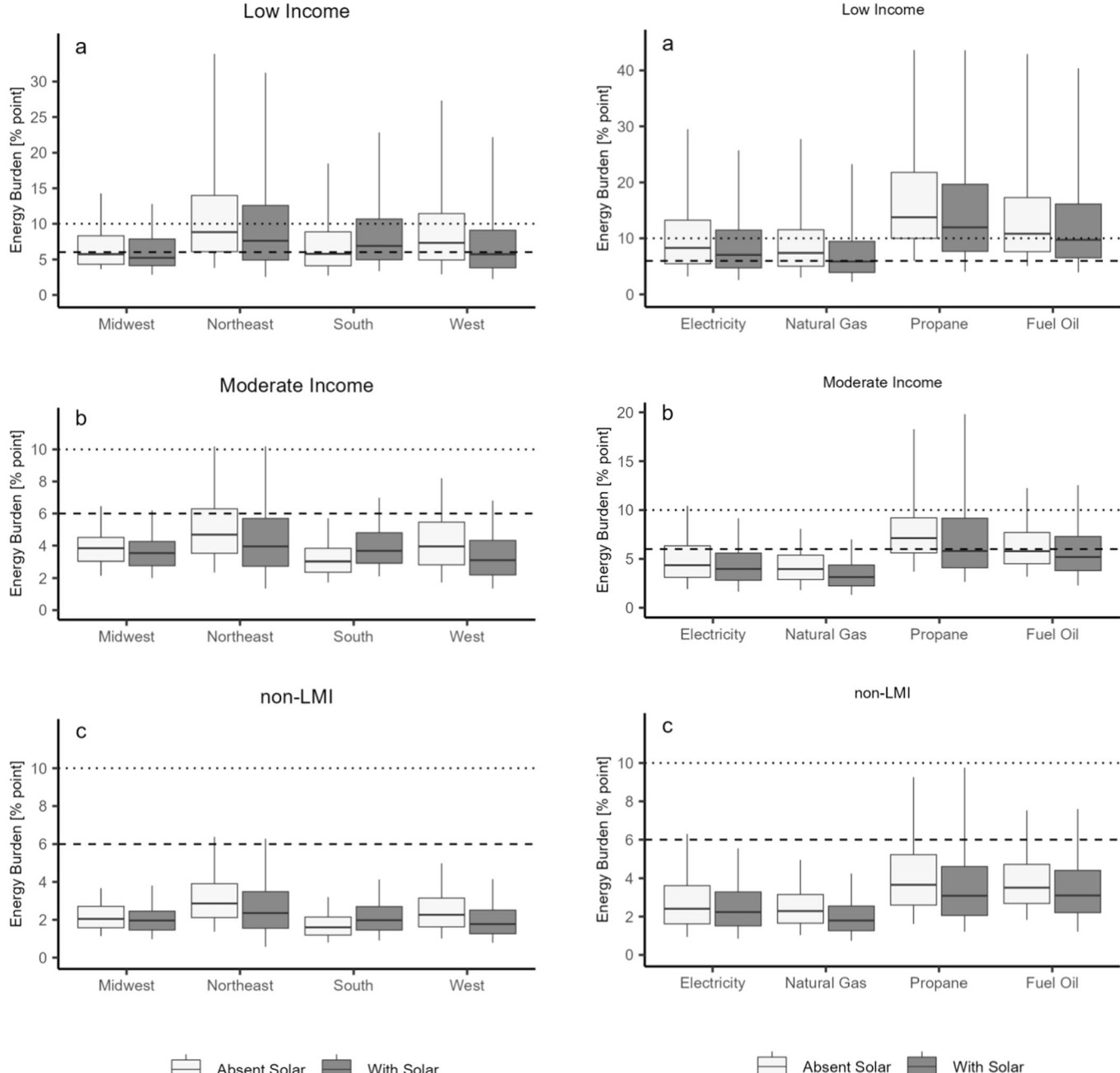

**Fig. 6 | Distribution of energy burden (EB) with and without solar across U.S. regions.** **a** Low-income (*N* = 95,215), **b** moderate-income (*N* = 87,388), and **c** non-Low- to moderate-income (non-LMI) (*N* = 239,063) adopters: Mid-line indicates median, lower (upper) box boundaries denote the 25th (75th) percentile, and lower (upper) whiskers are the 5th (95th) percentile. Dotted line at 6% reflects the cutoff for a high EB, and at 10% reflects the cutoff for a severe EB. Source data are provided with this paper.

**Fig. 7 | Distribution of energy burden (EB) with and without rooftop solar across household primary heating fuel type.** **a** Low-income (*N* = 95,215), **b** moderate-income (*N* = 87,388), and **c** non-low- to moderate-income (non-LMI) (*N* = 239,063) adopters: Mid-line indicates median, lower (upper) box boundaries denote the 25th (75th) percentile, and lower (upper) whiskers are the 5th (95th) percentile. Dotted line at 6% reflects the cutoff for a high EB, and at 10% reflects the cutoff for a severe EB. Source data are provided with this paper.

This could result in an overestimation of energy costs for those that ration usage as a coping strategy to hedge against high expected energy costs. Second, Experian income estimates were provided for 2021 annual household income (as opposed to at the time of adoption). This was done in order to capture true energy burden impacts during our study period, however, their proprietary model may include biases unknown to the researchers. Third, while we use the most likely hourly residential, retail electricity volumetric, and fixed tariff by zip code and presence of solar, we do not include specific low-income discount rates. This may result in overestimating EB for those

who are enrolled in these discounted tariffs. On a related note, empirical tariff data from 2021 does not capture more recent rate reforms such as default time of use rates or the movement away from net metering solar compensation. As such, this is an area for further exploration, especially for studies forecasting rooftop solar impacts on EB into the future. Another avenue of future work could expand this analysis to additional demographic elements.

Beyond methodological limitations, we acknowledge both EB's value as a proxy for affordability as well as its shortcomings. Namely, while it can help identify homes committing a high percentage of

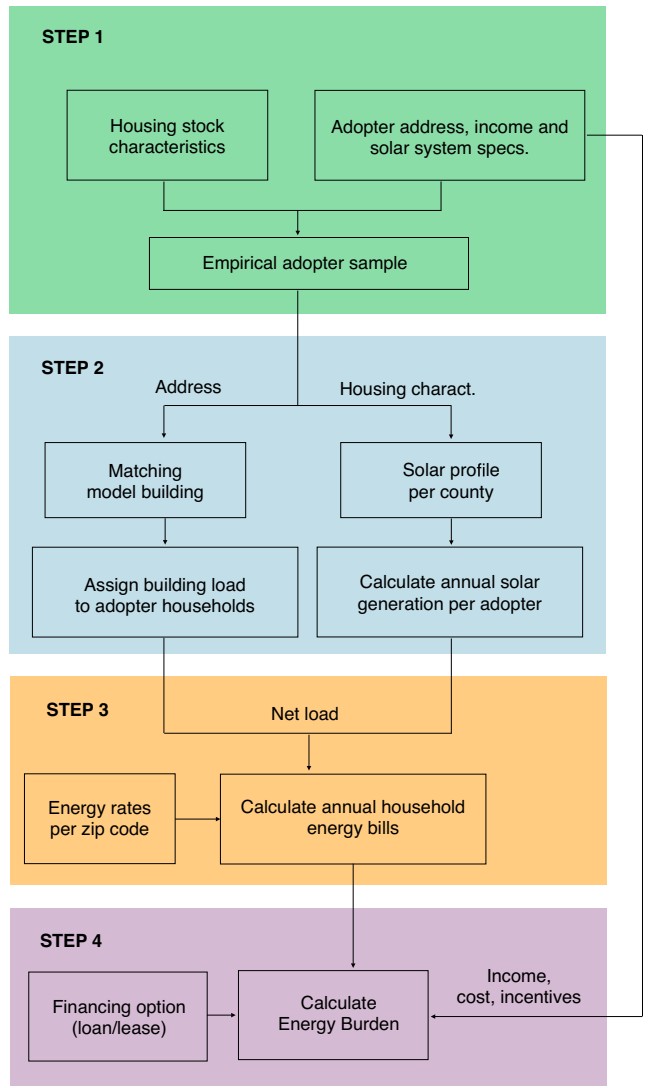

**Fig. 8 | Summary of steps used in this analysis.** Step 1 includes the filtering and merging of household data, Step 2 includes the estimation of hourly solar generation and household load, Step 3 includes the calculation of bills, and Step 4 integrates solar adoption costs. Source data are provided with this paper.

income to pay for energy services, it cannot capture households who may be energy rationing, such as reducing their bill by keeping their homes at unsafe or uncomfortable temperatures[10,36,37]. Moreover, it cannot speak to the underlying structural causes of energy insecurity or measure its non-economic effects, including physical and mental health as well as housing conditions and comfort[14,16,22,38–43]. Baker et al. (2023) cite various tradeoffs between equity metrics but state that they should be decision-relevant, understandable, and measurable, using data collected at a micro-spatial scale[10]. Notwithstanding the limitations of the energy burden metric, this paper leverages several sources of empirical and household-level data. Further, the increased use of EB as an indicator and evaluation metric in U.S. goals and policy at the state and federal levels has made it decision-relevant. Even so, future work could examine rooftop solar adoption's impact on other equity metrics.

Rooftop solar can support state and federal goals to reduce EB, including for LMI households. Nevertheless, there was a large fraction of low-income households whose post-adoption EB remained high (6–10%) or severe (over 10%), indicating persistent energy affordability issues. Notably, for low-income adopters with propane or fuel oil

heating, 80% maintained an EB above 6% after solar adoption. To alleviate this, policymakers may consider focusing low-income incentives on this group to reduce upfront costs, cost of capital, and/or penalties for non-payment. Policymakers may also wish to pair low-income solar incentives with additional interventions such as weatherization and/or fuel switching in order to reduce EB for those with a high percentage of non-electric energy costs.

## Methods

This study uses a four-step framework to analyze the financial impacts of rooftop solar adoption that occurred between 2013 and 2021 on EB at the household level in the year 2021, summarized in Fig. 8 and detailed through subsections below. It is important to note that the foundation of our data is an empirical dataset of actual solar adopters. In order to present our findings with and without solar, we follow these four steps to isolate the impact of solar adoption on household energy burden. First, we merge 2021 household income estimates, solar system attributes, and housing stock data for roughly 500,000 residential solar adopters across the U.S. (hereafter, "adopter households"). Second, we estimate hourly solar generation for each household based on its county and system size. To estimate hourly loads, we match each building to the simulated load profile of a building with similar characteristics. Third, we identify the applicable electricity rates and fuel prices for each adopter to calculate household energy bills. Fourth and finally, we calculate EB for the adopter households, considering incentives, upfront costs, and financing. The granularity and sources of the data for each of the steps in this modeling framework can be consulted in Supplementary Table 1. Institutional policy at Lawrence Berkeley National Laboratory at the time of the project's completion did not explicitly require IRB review for categories of data use applicable to this study.

### Filtering and merging household data

The analysis relies on Berkeley Lab's national dataset of residential solar adopters. The dataset version utilized for this study consists of roughly 2.5 million residential rooftop solar systems installed through 2021, representing 80% of all U.S. systems, and includes data on system size, cost, financing, and street address, among other items[25]. We append to this dataset estimated 2021 household incomes developed by Experian and housing stock data from CoreLogic.

We then filter this dataset to occupied single-family homes with rooftop solar installed on or after 2000 and installations ≤20 kilowatts (kW) with complete data on solar system size, cost, and estimated income. See Supplementary Table 2 for initial and final counts by state and Supplementary Fig. 1 for a map showing the geographic distribution.

### Estimating hourly solar generation and household load

To calculate the adopter household's net load, we separately estimate hourly solar generation and household energy consumption before rooftop solar installation.

We set out with a target of modeling 500k solar-adopting homes but aim to keep state-level geographical representation of the original set of solar adopter homes in the Solar Demographics dataset[25]. We hence specify the number of households to sample per state, such that the geographic distribution of the 500k homes is equal to the original distribution of homes.

In order to estimate the hourly load, we match each adopter household's property characteristics to a modeled building with similar housing characteristics (Table 2). The modeled building characteristics and calculated consumption profiles come from the National Renewable Energy Laboratory's End-Use Load Profiles[44]. Due to our need for hourly load data linked to geographic, income, and property data specific to 2021 and across a large number of U.S. states, it was not feasible to gather and use empirical end-use load data. As

**Table 2 | Variables matched between empirical and modeled property data with hourly end-use load profiles along with mean, standard deviation, minimum, and maximum of the final dataset's numeric variables**

| Variable | Variable type | Mean | Standard dev. | Range (min. – max.) |
|---|---|---|---|---|
| State | Categorical | N/A | N/A | N/A |
| Climate zone | Categorical | N/A | N/A | N/A |
| Heating fuel type | Categorical | N/A | N/A | N/A |
| Air conditioning type | Categorical | N/A | N/A | N/A |
| Pool indicator | Binary | 15% with pools | N/A | N/A |
| Number of persons | Numeric | 3.0 | 1.7 | 1–8 |
| Number of stories | Numeric | 1.4 | 0.49 | 1–4.5 |
| Square feet (living space) | Numeric | 1945 | 726 | 380–18,370 |
| Year built | Numeric | 1977 | 26 | 1900–2021 |

such, we opted for the best available source created and validated using empirical load data from 11 utilities and 2.3 million customer meters together with physical models ultimately representative of 133 million residences across the nation[45]. The model includes adjustments for location, empirical weather conditions, modeled stochastic occupant behavior, and more, resulting in an uncertainty range between 3% and 6% for annual load levels and between 1% and 4% for daily minimum base load[45].

We associate the modeled building's hourly load profiles with its matched adopter household. We then sample without replacement from this matched dataset until we reach the pre-determined state-level home counts, such that our final sample ($n = 500,010$) is geographically representative of the original dataset. For more details on this process, see Supplementary Methods and Supplementary Tables 3 and 4 in particular.

In order to estimate hourly solar production for adopter households, we use each household's county centroid to create an hourly profile with the National Renewable Energy Laboratory's System Advisor Model and then scale this based on respective, empirical installation size[46]. We assume a South-facing array, tilt at latitude, a fixed array with a 1.2 inverter loading ratio, and 14% system losses[47]. For system degradation, we assume 0.5% degradation per year from the year of installation to 2021, from the system's install date.

### Calculating energy bills

To calculate energy bills for each adopter household, we downloaded the electricity tariffs that residential customers were most likely to take service under in 2021 based on their zip code, with and without the presence of solar, using Genability software[48]. We then generated fixed [$] and volumetric [$/kWh] hourly electricity costs for each customer as they would be with and without rooftop solar. We assumed net metering was allowed for all adopter households (i.e., solar was compensated as negative load at the prevailing volumetric rate).

For non-electric energy costs, we used 2021 costs for natural gas, heating oil, and propane by state. With electricity rates and fuel costs together with the hourly net load calculated in Step 2, annual energy bills were calculated for adopter households. For more information on energy costs, see Supplementary Methods and Supplementary Table 5 in particular.

### Solar adoption costs and incentives

We use empirical data on each adopter's solar system costs, upfront incentives, and ownership structure to model different financing mechanisms available for adopter homes to pay for the upfront costs of rooftop solar installation. For third-party-owned systems, we assume a lease structure in which any tax credits or ownership-based benefits go to the third-party, which may, in turn, pass along these financial benefits to the customer in the form of a lower lease rate[49,50]. Since this analysis focuses on the impacts of off-bill repayment of solar

systems, we assume that host-owned systems are loan financed, as opposed to an upfront cash purchase (see Supplementary Methods and Supplementary Fig. 3 and Supplementary Table 6 in particular). In this case, any upfront incentives, performance-based incentives, investment tax credit(s), and solar renewable energy credits—flow directly to the owner (see Supplementary Methods and Supplementary Tables 7–9 in particular). While we do not focus on systems paid for upfront with cash, we do include a comparison between loans and upfront cash payments via net present value calculations and find similar values, which indicates that the financial model for the loan would look similar to the financial model for an upfront payment (see Supplementary Figs. 2 and 4). Since the available loan and lease data only goes back to 2013, we henceforth concentrate our analysis on the adopter homes that installed solar after 2013, representing 422k adopters.

Finally, to calculate EB for the study period of 2021, the adopter household's income estimates, energy bills, off-bill costs, and revenue are combined to quantify the comprehensive financial impact of rooftop solar adoption (Eq. (1)).

$$EB[\%] = \frac{\text{Energy Bills}_{postPV}[\$] - \text{PV incentive}_{\$/kWh}[\$] + \text{PV Repayment}_{loan.lease}[\$]}{\text{Annual Gross Income}[\$]} \quad (1)$$

### Reporting summary

Further information on research design is available in the Nature Portfolio Reporting Summary linked to this article.

## Data availability

The datasets generated during the current study are available in the figshare repository, https://doi.org/10.6084/m9.figshare.25130498 in the Source Data file. Additional data that support the findings of this study, such as aggregated results, are available from the corresponding author upon request. The raw household-specific data (e.g., individual property information, household-level hourly electricity residential, retail tariff, individual solar installation information, household income estimates, and personally identifiable information) are protected and not available due to data privacy laws as they were obtained under non-disclosure agreements or via publicly-available paid subscriptions. Source data are provided with this paper.

## Code availability

Code generated to create the figures is available at https://doi.org/10.5281/zenodo.11089676. Additional code is available upon request.

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

## Acknowledgements
This work was supported by the U.S. Department of Energy's Office of Energy Efficiency and Renewable Energy under Solar Energy Technologies Office Agreement Number 388444 and Contract No. DE-AC02-05CH11231.

## Author contributions
S.P.F. conceived and designed the analysis, collected the data, contributed data or analysis tools, performed the analysis, wrote the paper. C.C.M. provided feedback on scope, collected the data, contributed data or analysis tools, performed the analysis, and provided edits. E.O. provided feedback on scope, contributed data or analysis tools, performed the analysis, and provided edits. G.B. provided feedback on scope, collected the data.

## Competing interests
The authors declare no competing interests.
