## [Peer Review File · Nature Communications]

REVIEWER COMMENTS

Reviewer #1 (Remarks to the Author):

The work presented represents a novel use of data sets and approaches to explore the role of solar PV in reducing energy burden in the US, which is a very timely topic of relevance to a large audience.

The extensive methodology provides a view of the thorough work the authors have done, and at the same time raises concern to the validity of the final outcomes presented based on the assumptions made and expressed limitations.

- For example, there are probabilities made from the Residential Energy Consumption Survey and applied to the household level sample that are unclear on how were they made and could benefit from further explanation.

- The authors recognize early on in the article that shortcomings of EB. However, when conducting the analysis, it seems income/race/and more are excluded which would drive significant variations in the final modeled fuel use curves, likely influencing the results/conclusions presented. Further clarification of the impact those omissions might have in the results would be appreciated.

- As acknowledged in the Conclusion section, there are significant limitations regarding the use of synthetic data. Overall, there is a very strong the solar data analysis (electricity generation), but more evidence seems needed to provide certainty in the consumption (electricity demand) for this type of analysis (for example, through a solid proof/validation of the synthetic data and real building-energy relation?). These points represent major assumptions on the study, which has a high degree of speculation that could be potentially addressed with real electricity consumption data either from utilities, or alternatively through more rigorous sensitivity analyses.

Minor comment/question:

- Can you elaborate why both \$/W and income have the same coefficient sign? (Table SI.9)

Reviewer #2 (Remarks to the Author):

The paper titled "Effects of Rooftop Solar on Household Energy Burden in the United States" examines how rooftop solar installations affect the energy burden (EB) of U.S. households. The study used a dataset covering different regions of the U.S. and hourly energy demand curves, showing that rooftop solar generally reduces the energy burden for most adopters. It not only compares low- and middle-

income households but also includes Off-Bill Financial Impacts, which is interesting. However, the paper lacks statistical rigor for Nature Communications. While the analysis is interesting, it is too descriptive in its present form without robustness checks. The existing findings from the descriptive analysis are “intuitive” and the innovations and novelty of the study is significantly limited by the lack of statistically significant differences between different groups. For instance, how much energy burdens can be attributable to rooftop solar adoption instead of other influencing factors? Further statistical quantification of the pure effects of residential solar adoption is needed, rather than just comparing EB values across adopters & non-adopters for different income groups. Local electricity tariffs, rate structures, and housing conditions can all affect energy burden dramatically. It is also unclear how different state and local policies affect and deter solar rooftop effects on household energy burdens. For instance, in locations where monthly solar demand charges exist, this could limit the effect of solar adoption on household energy burden. In locations where net metering programs exist or self-consumption is encouraged, it may be challenging to address the counterfactual potential savings. This study does take a look at the national dataset, but perhaps some specific statewide analyses would be more beneficial first due to the heterogeneity across local utilities and states in their solar rooftop barriers and incentives. More detailed and interesting findings should be detected using more quantitative analyses. Additionally, it seems that all final datasets are aggregated at a state-level, if so you should report your spatial resolution as this is not clear from the current version of the manuscript. To address Justice40 initiatives, a higher resolution is needed since they are identified at a sub-state level resolution.

The literature review on energy burden studies is relatively limited as there are now many reviews on alternative metrics to energy burden for affordability and other studies that have been neglected. For further description of established energy burden literature, consider the following review that captures many other studies in this area:

Baker, Erin, Sanya Carley, Sergio Castellanos, Destenie Nock, Joe F. Bozeman III, David Konisky, Chukwuka G. Monyei, Monisha Shah, and Benjamin Sovacool. "Metrics for Decision-Making in Energy Justice." *Annual Review of Environment and Resources* 48 (2023): 737-760.

How does the cost of financing for a solar loan affect the outcome on energy burden? It is not clear the assumptions such as interest rate and cost of leasing.

How are climate zones taken into consideration into the simulated load profiles - since the resolution of the model is not clear, it is unclear whether the loads are the same for different regions, states, or just building types scaled at a national level, which would potentially ignore climate effects on household energy use.

Solar profile is at a county level, but what about homes that cannot support solar due to local shading or roof issues? Could they still reduce their energy burden and how does this study demonstrate that?

Not enough details in the methodology section are reported to replicate the study as there appear to be flaws when interpreting the statistical results. A decomposition analysis is recommended.

Specific comments:

On page 4 line 114, I doubt the effectiveness of using linear regression to capture the relationship between these potential influencing factors. Have you conducted any statistical tests to prove your choice of model? Did you compare other models or find some ways to prove the robustness of your regression results?

On page 4 line 117, What do you mean "in that order of magnitude"? Do you mean the numbers of coefficients? If your independent variables have different units then the coefficients are not comparable. You can use decomposition methods to find the exact contribution of each factor.

On page 4 line 199, could you also compare energy burden differences for households with different housing conditions?

On page 6 line 148, Section 2.3 Energy burden impacts in 2021 based on year of adoption. It seems that you are using 2021 energy prices, have you converted 2013-2020 income to the 2021 price level for comparison to account for inflationary changes?

On page 6 line 161, section 2.4 Regional and state differences. You mention regional differences, but keep referring to SI 7. I suggest conducting regressions by regions to prove "regional difference" instead of concluding your results from overall regression. This is misapplying the downscaling of the statistical results.

On page 13 line 270, Table 1: Variables matched between empirical and modeled property data with hourly end-use load profiles. You should also report variable statistics such as mean, sd, min, and max instead of only reporting variable types.

Reviewer #3 (Remarks to the Author):

I appreciate the study's valuable contribution to the body of knowledge, highlighting the need of integrating energy efficiency and electrification measures alongside rooftop solar adoption to mitigate energy burdens. The study is both intriguing and highly original. However, I have some specific points to address:

1. Page 2, First Paragraph:

Providing supporting references in the first paragraph on page 2 would bolster the credibility of the argument that the burden of air pollution and cost shifts is uneven.

2. Page 2, Second Paragraph:

The examples provided (low-income households using fuel oil, African American households in mixed-humid climate zones, etc.) seem to lack a direct connection with health impacts and are more related to socioeconomic, housing, and built environment characteristics. Suggest removing the term "health impacts" from the sentence as the health impacts were introduced in the subsequent sentences.

3. Page 2, Third Paragraph:

While it is commendable to see the limitation of the definition "energy burden" and the rationale for focusing on this topic, I recommend moving this information to the study's limitations section, as it aligns more with future research opportunities.

4. Page 3, First Paragraph:

Providing supporting references for the argument that existing literature on energy burden reduction from solar focuses solely on bill impacts, without accounting for ongoing off-bill financial impacts of solar adoption, would strengthen the narrative. If available, comparing these references with the study's results could emphasize the significance of the current study.

5. Figure 1 - Lease Option:

The assumption that all solar adoption credit benefits go to the third party, leading to reduced prices for customers in the form of lower lease rates, may not always hold true. A sensitivity analysis, considering the magnitude of credit pass-through, would be valuable for comparing impacts between lease and loan options. This is crucial as the results for the lease option could potentially be underestimated.

6. Figure 5:

It would be beneficial to explain the decreasing pattern of energy burden with solar from 2020 to 2021 in Figure 5, as this trend is quite apparent. Providing context for this observation would enhance the reader's understanding.

I hope these suggestions contribute to the further refinement of your study.

Reviewer 1

1. There are probabilities made from the Residential Energy Consumption Survey and applied to the household level sample that are unclear on how were they made and could benefit from further explanation.

We have included a new paragraph (Supplemental Information 3 “Matching adopter housing characteristics to modeled buildings”, paragraph 4) that details both our thought process and steps to generate air conditioning estimates. This includes new citations that detail the method used to generate relative standard error used to compare RECS groupings as well as citations supporting evidence that air conditioning adoption is dependent both on climate (location) and income, which was the basis of why we derived estimates from grouping RECS respondents by state and income group. We hope that this adds sufficient clarity. Please see addition copied and pasted below:

“For air conditioning type, we used RECS to calculate probabilities across four categories (None, Central, Room, Heat Pump) based on state and income group. This involved grouping the RECS microdata by state and income (aggregated into two groups over and under \$100,000 annual income) and assessing the relative standard errors (RSE) in R as per RECS guidance. Since air conditioner presence and type are dependent both on location (climate) and income, it was important to generate estimates on both of those indicators while keeping RSE as low as possible, so we analyzed RSE for variations of income groupings (from the available 16 income groups to 2) and spatial groupings (from four regions to every state). Higher aggregation and lower granularity yields lower RSE. Increasing granularity to state level did not produce a high number of large RSE values, however, income groupings introduced some large values across specific AC categories. Nevertheless, RSE remained at lower levels when income was aggregated into two groups (over and under \$100,000), which led to our final selection. With these groupings by state and two income groups, the cases where RSE exceeded 30% were only in locations where estimates indicated low prevalence (likely due to limited sample size). Specifically, across each and every single state/income grouping and AC category, the largest estimate where RSE was higher than 30% was 11% prevalence, which is relatively low. In these specific state/income cases, other AC categories had sufficiently low RSE, so the overall distribution of AC groups was acceptable. In the end, balancing the tradeoffs of RSE versus grouping granularity allowed us to produce likelihood for each adopter to have no AC, central AC, portable/window AC, or heat pump AC based on their state and income. These probabilities, in turn, informed random selections for each household, which was then used for adopters with missing air conditioning values from CoreLogic.”

2. The authors recognize early on in the article that shortcomings of EB. However, when conducting the analysis, it seems income/race/and more are excluded which would drive significant variations in the final modeled fuel use curves, likely influencing the results/conclusions presented. Further clarification of the impact those omissions might have in the results would be appreciated.

We apologize that we may not fully understand this comment, as the connection between the first sentence (shortcomings of EB) and the second sentence (demographic factors in our analysis) is not clear. We believe the reviewer is noting how other demographic factors may influence the impacts of solar on energy burden. If so, we fully agree, though this is a large question that is outside the scope of our paper.

We nodded to further exploration of other demographic factors in the revised manuscript's Discussion section and added two citations (end of paragraph 4):

"Another avenue of exploration could extend this analysis to other demographic variables. For instance, research shows that energy use intensity varies with demographic variables such as race and ethnicity. As such, rooftop solar may differentially affect energy burden across demographic factors other than income, such as race, rurality, and education. Analysis of the interaction between solar adoption, energy burden, and adopter demographics is an area for further research."

3. As acknowledged in the Conclusion section, there are significant limitations regarding the use of synthetic data. Overall, there is a very strong the solar data analysis (electricity generation), but more evidence seems needed to provide certainty in the consumption (electricity demand) for this type of analysis (for example, through a solid proof/validation of the synthetic data and real building-energy relation?). These points represent major assumptions on the study, which has a high degree of speculation that could be potentially addressed with real electricity consumption data either from utilities, or alternatively through more rigorous sensitivity analyses.

In order to maintain comparable and consistent national coverage of our results in a standardized manner, we used modeled synthetic data. While LBNL has access to some hourly metered data, these data were collected roughly a decade ago and the only other metadata available are utility territory and retail rate structure. Since this analysis relies on concurrent income, location, solar installation data, and property characteristics (most of which is current and empirical, and all of which is specific to an address), the only feasible way to perform this analysis was to use modeled data, and so we used the best available. That said, we agree with Reviewer 1 that we could provide better information on the models themselves. As it stands, the End Use Load Profiles (and ResStock from which they were generated) are considered the best available in this category. It is understandable that a reader not familiar with this model may be justifiably skeptical of the assumptions. As such, we have made edits to include a citation to NREL's technical report outlining their assumptions, calibration, validation, and uncertainty. In addition, we have added language that highlights some main critical points to improve trust in the source. We felt that it was important to include this in the main body of the paper as opposed to the SI, so the justification for using modeled data, the new citation, and the main points on the model can be found in Section 4.2 ("Estimating hourly solar generation and household load"), paragraph 3:

"Due to our need for hourly load data linked to geographic, income, and property data specific to 2021 and across a large number of U.S. states, it was not feasible to gather and use empirical end use load data. As such, we opted for the best available source created using empirical load data from 11 utilities and 2.3 million customer meters together with physical models ultimately representative of 133 million residences across the nation. The model includes adjustments for location, empirical weather conditions, modeled stochastic occupant behavior, and more resulting in an uncertainty range between 3% and 6% for annual load levels and between 1% and 4% for daily minimum base load."

We also edited a sentence in the fourth paragraph of the Discussion to now read "While this analysis was only possible with the use of synthetic load data, this can propagate any biases...." to clarify the need to use these data as well as the limitations that the use of it presents.

4. Can you elaborate why both \$/W and income have the same coefficient sign? (Table SI.10)

In Supplemental Table 10 the coefficients of $\$/W$ and Income are negative (-0.35 and -0.08, respectively). This means that as both the unit cost of solar and the household income go up, the level of savings that solar provides goes down (as a difference in energy burden). Both have p-values near 0 so are both significant, however, the unit cost of solar's magnitude is over four times that of income. This is intuitive as more expensive unit cost of solar may be associated with a higher level of monthly repayment for a lower level of bill savings. That of income is less important, but indicates that a higher income is associated with lower energy burden savings. This may be simply because income is a denominator of energy burden, thus, the same reduction in bills would lead to a muted energy impact reduction for one household with a higher income than the other. Additionally, solar adopters have are not always motivated by economics (especially in earlier years of adoption, but even more recently as demonstrated in the Agdas & Barooah that we cite), so conceivably, those who adopt solar for reasons other than economics (i.e., those that see smaller energy burden reduction or those that may actually see an increase in energy burden) may be more likely to be higher income. Without further exploration of this specific question, we are hesitant to make any assumptions and only state the correlation here in Supplemental Table 10 to help describe an approximate ranking of the importance of different factors in explaining changes in EB in our model.

Reviewer 2

1. While the analysis is interesting, it is too descriptive in its present form without robustness checks. The existing findings from the descriptive analysis are “intuitive” and the innovations and novelty of the study is significantly limited by the lack of statistically significant differences between different groups. For instance, how much energy burdens can be attributable to rooftop solar adoption instead of other influencing factors? Further statistical quantification of the pure effects of residential solar adoption is needed, rather than just comparing EB values across adopters & non-adopters for different income groups.

We first offer two very important clarifications, but then re-examine this comment and try to apply the spirit of the comment to our revision of this paper. Reviewer 2 states "Further statistical quantification of the pure effects of residential solar adoption is needed, rather than just comparing EB values across adopters & non-adopters for different income groups." It is first very important to clarify that we are not comparing EB values across adopters and non-adopters. Each and every household in this analysis is a solar adopter for which we have empirical solar installation information. As such, we model the "pure" effect of residential solar by isolating and removing the solar installation's impact on modeled household electricity bills. Thus, we model the distribution of energy burden with and without solar. The difference between those distributions is, by our construction of the model, the “pure” effect of solar. This allows us to attribute all of the EB reduction to solar alone (since everything else is held static) instead of other influencing factors, which is the second important clarification. Thus, all changes in EB are attributable completely to rooftop solar (by design).

With those clarifications, we still want to address this comment and we re-interpret the comment here to suggest that:

First, we must make it more clear that these are all rooftop solar adopters and we have already isolated the impact of solar on energy burden and electricity tariff assignment, by design. We have added this explicitly as the second sentence of our Methods section, which now reads “It is important to note that

the foundation of our data is an empirical dataset of solar adopters. In order to present our findings “with” and “without” solar, we follow these four steps to model the impact of solar adoption on household energy burden.”. Moreover, even though the first sentence in Methods “Calculating energy bills” already conveys that the retail rate that the customer is on depends on both zip code and presence of solar, we re-iterate this by adding additional information in Supplemental Information 4 “electricity rates and fuel costs” where we have added information in the first paragraph to show how customers’ rates were impacted by solar adoption: “To calculate energy bills for the adopter households, we downloaded the electricity tariffs that residential customers were most likely to have in 2021 based on their zip code with and without presence of solar, using Genability software. Of the final set of customers, only 38% of tariffs were the same with and without solar while a majority were moved to different rates. Additionally, absent solar, 6.2% of our sample were on time-sensitive rates. However, with solar adoption this increased to 33.2%.”

Second, even though our methods already isolate the impact of solar on energy burden, we agree with Reviewer 2 that this analysis could benefit from additional statistical analyses to strengthen our argument. As such, we have added Supplemental Information 8 “Statistics” where we describe the ANOVA and t-tests used to test for differences in the impacts of PV on energy burden across sub-populations (e.g., low vs. high income) in our results section. We also describe an additional Monte Carlo analysis where we test how robust our results are to variations across three key variables. We find that our results across groups are statistically different from one another and that our results are robust across variation of key variables. Specifically, we find that energy burden reduction due to solar is particularly strong among low-income adopters and that there are significant differences across regions.

Third, the reviewer refers to a lack of statistically significant differences “between different groups.” The intent of this comment is not clear, but we believe the reviewer refers to our conclusions about how solar reduces energy burden across sample subsets (e.g., income, region). These differences are mostly statistically significant, it was simply our oversight not to report comparative statistics in the original manuscript. In addition to creating Supplemental Information 8 “Statistics” that details the statistical tests to demonstrate differences between groups (ANOVA and t-tests). In addition, we have added a paragraph at the beginning of “Results” paired with a figure that shows the average EB reduction across groups due to solar with respective 95% confidence intervals.

Please see the added text at the top of the Results section; the added text dispersed throughout the “Results” section (locations cited); and the new Supplemental Information 8 “Statistics” that includes Supplemental Figure 5 with the results of the Monte Carlo analysis copied and pasted at the bottom of this response.

Please find the added paragraph and figure at the top of “Results” copied and pasted here:

“Energy burden impacts of rooftop solar were analyzed across ownership models, income group, year of solar adoption, region, and heating fuel type. Figure 1 shows the average change in EB across each group’s population with respective 95% confidence intervals. This illustrates the differential impact of these factors on EB reduction. We provide further detail on these findings in the subsections below, and include results from ANOVA, t-tests, and Monte Carlo analyses throughout this section as well as in Supplemental Information 8 “Statistics” that demonstrate the robustness of our results to variation and that each group is statistically different from one another.”

Figure 1: Mean energy burden percentage point reduction due to rooftop solar adoption for households, across groups, with error bars showing the 95% confidence intervals.

Next, please find the added text dispersed throughout the “Results” section (sub-sections and paragraphs cited) copied and pasted here:

“On average, solar adoption reduced household energy burdens for systems financed through lease payments by roughly 0.2 percentage points more than for systems financed through loans ($t=-38.3$).” (“Results” > “Energy burden impacts of rooftop solar in 2021 based on ownership model” paragraph 1)

“Solar adoption reduced low-income household energy burden by roughly 1.3 percentage points more than for high-income households ($F=15061.9$, $p<0.0005$).” (“Results” > “Energy burden impacts of rooftop solar in 2021 by income group” paragraph 2)

“In all, solar reduced EB for households in the West roughly 1.6 percentage points more than for households in the South (F=2132.1, p<0.0005).” (“Results” > “Energy burden impacts of rooftop solar in 2021 across regions” paragraph 2)

“Solar adoption reduced EB for households using natural gas as a primary heating fuel by roughly 0.3 percentage points more than those using fuel oil (F=157.2, p<0.0005).” (“Results” > “Energy burden impacts of rooftop solar in 2021 across primary heating fuel types” paragraph 1)

Third and finally, please find the added Supplemental Information 8 “Statistics” copied and pasted here:

“To test whether each population across the groups of interest (i.e., ownership model, income, year of adoption, region, and heating fuel type) was statistically different in terms of their energy burden reduction due to solar and to test how robust our results were to changes in inputs, we conducted several statistical tests. To address the former, we conducted ANOVA and t-tests, and to address the latter we conducted a Monte Carlo detailed below. To summarize, we found that the energy burden reduction of each population in their respective groups were statistically different. We also found our results to be robust over changes to lease terms, loan terms, and rooftop solar shading and soiling.

Independent sample t-test (used when comparing two groups) and one way ANOVA tests (used when comparing more than two groups) showed that energy burden reductions were statistically different across groups. Tests were implemented using Python package scipy.stats. On average, solar reduced household EB for systems financed through lease payments (n=187087) by roughly 0.2 percentage points more than for systems financed through loans (n=234579) (t=-38.3, p<0.0005, DF=413415.7). Solar adoption reduced low-income household EB by roughly 1.3 percentage points more than for high-income households (F=15061.9, p<0.0005, n=421666, df1=2, df2=421663). Solar adoption reduced EB for households using natural gas as a primary heating fuel by roughly 0.3 percentage points more than those using fuel oil (F=157.2, p<0.0005, n=421666, df1=3, df2=421662). Finally, solar adoption reduced EB for households in the West roughly 1.6 percentage points more than for households in the South (F=2132.1, p<0.0005, n=421666, df1=3, df2=421662).

In addition to those comparative statistics, we ran a series of Monte Carlo simulations allowing key inputs to vary over 1,000 simulations. We did not allow any variation of empirical data (i.e., rooftop solar installation cost, size, incentives; location; property information at the household level or hourly retail tariff empirical at the zip code level) or values calibrated with empirical data (i.e., load curves matched by empirical property information and calibrated by ratio of solar generation to average annual load). Instead, we focused on allowing variation for the data that was either modeled (i.e., solar generation by county centroid) or aggregated at a less granular level (i.e., financial assumptions aggregated from empirical data to respective state/year combinations). In sum, we explore variations in three key parameters: the interest rate for loans, monthly lease payment values, and the soiling losses of rooftop solar systems.

We generate distributions for the financial parameters (interest rate for loans and monthly lease payment values) using values from historic solar quotes available from EnergySage. Each bootstrap iteration includes an interest rate and a monthly lease payment value that is constructed from: 1) the deterministic value described in Supplemental Information 5 “Financing mechanisms for rooftop solar” (calculated based on median values for these parameters per year and state in the EnergySage data) and

2) a stochastic adjustment that is randomly drawn normal distributions with a mean of 0 and a standard deviation equivalent to that observed across all solar loan or lease quotes available for that year in the EnergySage dataset, as applicable (see Supplemental Table 6). The result is that each adopter's interest rate or monthly lease payment is selected randomly for each of the 1,000 iterations from a normal distribution where the mean is set by each adopter's deterministic value and the standard deviation depends on the year of adoption and ownership structure.

In order to stochastically adjust the rooftop solar production of our main results, we modify the deterministic potential rooftop solar production for the centroid of the county where each household is located by applying a stochastic soiling loss factor that is sampled randomly from a normal distribution of soiling loss factors from NREL's Photovoltaic Soiling Map with mean 0.988 and standard deviation 0.015.50 The soiling factor randomly sampled is capped at 1.

We run 1,000 bootstrap iterations and record the distribution of energy burden with and without solar for each adopter. Our primary goal in running those Monte Carlo simulations was to develop confidence intervals based on the 5th and 95th percentile values from the simulations. Comparisons of the outputs of those simulations across groups provides a robustness check on some key findings in the manuscript. Specifically, the Monte Carlo simulations reiterate that EB reductions are significantly stronger among low-income households and that EB reductions vary significantly across regions, as illustrated in Supplemental Figure 5.

Supplemental Figure 5: Monte Carlo results: Distribution of median energy burden reductions due to solar adoption across the groups analyzed in this paper (ownership structure, income group, region, and heating fuel type). Points represent the average values across all simulations while the error bars illustrate the 5th and 95th percentile of the distributions over all 1,000 iterations

”

2. Local electricity tariffs, rate structures, and housing conditions can all affect energy burden dramatically. It is also unclear how different state and local policies affect and deter solar rooftop effects on household energy burdens. For instance, in locations where monthly solar demand charges exist, this could limit the effect of solar adoption on household energy burden. In locations where net metering programs exist or self-consumption is encouraged, it may be challenging to address the counterfactual potential savings. This study does take a look at the national dataset, but perhaps some specific statewide analyses would be more beneficial first due to the heterogeneity across local utilities and states in their solar rooftop barriers and incentives. More detailed and interesting findings should be detected using more quantitative analyses. Additionally, it seems that all final datasets are aggregated at a state-level, if so you should report your spatial resolution as this is not clear from the current version of the manuscript. To address Justice40 initiatives, a higher resolution is needed since they are identified at a sub-state level resolution.

We believe that there may have been a misunderstanding here. Even so, we do make edits to make sure that this is more clear and to add more context in the manuscript and supplemental information. To clarify, we mention granularity of the data several times in the body of the manuscript as well as lay it out explicitly in Supplemental Table 1 where each and every data point is explained along with the granularity, source, etc. We also point out in several locations that this is for 2021 with income estimates from 2021 (regardless of when adoption occurred), rate structures from 2021, solar resource from 2021, etc. The costs and incentives are empirical and thus are taken as-is, inflated to 2021 values. Further, in Methods Section “Solar adoption costs and incentives” we mention that we have empirical data of incentives received from adopters. As such, while we agree that there is heterogeneity across utilities and states for incentives- we believe that these are adequately captured in the empirical data. We also summarize additional state incentives in the form of tax credits and renewable energy credits in Supplemental Information 6 (Supplemental Tables 7, 8, and 9).

On the topic of tariffs and rate structures, we would like to point out that both “Methods” and Supplemental Tables 1 and 4 already point out that the tariff/rate structure are specific to the customer's zip code and presence of solar and include hourly fixed and volumetric values (the most “local” possible without having empirical household data). Further, Supplemental Information 4 is entirely dedicated to our methods surrounding electricity rates. While this did not result in any changes to our model, we made several edits to ensure that this is more clear to a reader:

We now explicitly point out that the granularity of the data can be found in Supplemental Table 1. In addition, paragraph 1 of “Methods” now reads “The granularity and sources of the data for each of the steps in this modeling framework can be consulted in Supplementary Information “Data Sources”.” We later reference Supplemental Information 4 for more information on rates where we greatly expanded the first paragraph to include the number of customers that changed rates upon solar adoption (roughly 62% of the sample) as well as the number of customers on time-of-use rates without solar (6%) and with solar (33%). Further, we remind the reader that these are empirical data from 2021, which is prior to

many current debates about net metering reform and indeed even before customers were moved from grandfathered flat rates to default time-of-use rates in states like California, etc. To even further emphasize this point and to highlight the fact that the scope of this paper leverages the use of historic, empirical data, we add an additional edit to our limitations section in the Discussion that re-iterates the focus on 2021 rates and the opportunity for further research on the impacts of rate reform on these results: "On a related note, empirical tariff data from 2021 does not capture recent rate reforms such as default time of use rates or the movement away from net metering solar compensation. As such, this is an area for further exploration, especially for studies forecasting rooftop solar impacts on EB into the future."

3. The literature review on energy burden studies is relatively limited as there are now many reviews on alternative metrics to energy burden for affordability and other studies that have been neglected. For further description of established energy burden literature, consider the following review that captures many other studies in this area: Baker, Erin, Sanya Carley, Sergio Castellanos, Destenie Nock, Joe F. Bozeman III, David Konisky, Chukwuka G. Monyei, Monisha Shah, and Benjamin Sovacool. "Metrics for Decision-Making in Energy Justice." *Annual Review of Environment and Resources* 48 (2023): 737-760.

The Baker et al. paper came out after the initial submission of this paper. We have added this citation in multiple locations throughout the introduction such as:

- *citation for rooftop solar and other energy technologies being distributed unevenly across the US (Introduction, paragraph 1)*
- *energy burden being the most commonly-used affordability metric in literature (Introduction, paragraph 1)*
- *limitations of energy burden as a metric (Discussion, paragraph 5- note that this section used to be in the Introduction, but was moved to address a comment by Reviewer 3)*
- *specific citation of the tradeoffs in metric selection and requirements of a stronger equity metric "Baker et al. (2023) cites various tradeoffs between equity metrics, but states that they should be "decision-relevant", understandable, and measurable, using data collected at a micro-spatial scale" (Discussion, paragraph 5- note that this section used to be in the Introduction, but was moved to address a comment by Reviewer 3)*
- *few studies leverage micro-data in studies on this topic (paragraph 4)*

In addition to Baker et al. (2023), we added additional citations on metrics, energy burden and other studies to address this comment including:

- *Barlow et al. (2022), which surveys metrics being used in practice and the relevant stakeholders (Introduction, paragraph 1)*
- *Scheier & Kittner (2022), which finds higher burdens across minority households (Introduction, paragraph 2)*

- Hardy et al. (2024), which is a new study that looks at energy burden impacts of solar among other interventions, but leaves gaps that this paper helps to fill. For example, Hardy et al. only looks at three locations, relies more heavily on modeled data, lacks income data, assumes cash payment upfront for solar system, and assigns uniform (i.e., does not vary) tariffs and solar system installation sizes to each household (Introduction, paragraph 4).

4. How does the cost of financing for a solar loan affect the outcome on energy burden? It is not clear the assumptions such as interest rate and cost of leasing.

In addition to our Monte Carlo simulations where we allowed three factors to vary, we ran separate simulations allowing only the lease and loan assumptions to vary. These simulations allow us to isolate the sensitivity of the impacts on EB to underlying assumptions around leasing and loans. The results of those simulations are depicted in Supplemental Figure 4. That figure suggests that the results are generally more sensitive to our leasing assumptions, while varying loan assumptions within reasonable bounds has relatively little impact on the results. In the main text, we reference this finding and where to go in the Supplemental Information in the second paragraph of Results Section “Energy burden impacts of rooftop solar in 2021 based on ownership model”. See new text, table, and figure copied and pasted below:

“In order to quantify and isolate the sensitivity of EB impacts to lease and loan terms, we ran 1,000 iterations allowing the lease assumptions alone to vary and an additional 1,000 iterations allowing the loan assumptions alone to vary (Supplementary Figure 4). Each bootstrap iteration randomly draws an interest rate or a monthly lease payment value from normal distributions with a mean of 0 and a standard deviation equivalent to that observed across all loan or lease quotes available for that year (see Supplementary Table 6). This stochastic adjustment is applied to each customer’s respective loan or lease terms.

Supplementary Table 6: Mean and standard deviation of loan interest rates and monthly lease payments per installed capacity, by year. The means are provided as additional information while the standard deviations are the values used to vary (as per a normally distributed set of values) each adopter's deterministic value over the 1,000 iterations, based on year of adoption

Year	Interest rate loans (%)		Monthly lease payments (\$/kW)	
	Mean	Standard Deviation	Mean	Standard Deviation
2013	4.398085	1.667746	11.8045	3.63065
2014	4.953258	1.597751	16.69708	3.602215
2015	5.202252	1.400885	14.95803	4.489026
2016	4.69869	1.335018	15.09683	4.617652
2017	4.459123	1.49145	16.05853	4.631676
2018	4.588306	1.332365	15.71183	3.655411
2019	4.470039	1.129354	17.84004	2.929264
2020	4.028624	1.317391	11.93169	3.607794
2021	2.97979	1.600502	13.43706	4.458521
2022	2.800157	1.628765	13.92194	4.249917
2023	4.493112	1.26838	13.04131	4.167668

Supplementary Figure 4: Median percentage point change in energy burden reduction across 1,000 iterations of varying lease terms (left) and across 1,000 iterations of varying loan terms (right). Boxes indicate inter-quartile ranges i.e., median, 25th, and 75th percentiles. Error bars depict 5th and 95th percentiles, and points depict outliers.

Supplementary Figure 4 indicates that our results are more sensitive to changes in lease terms than for changes in loan terms. For instance, the interquartile range of EB reductions across all adopters spans 5.8 percentage points when allowing lease terms to vary, compared to 2.8 points when allowing loan terms to vary.”

5. How are climate zones taken into consideration into the simulated load profiles - since the resolution of the model is not clear, it is unclear whether the loads are the same for different regions, states, or just building types scaled at a national level, which would potentially ignore climate effects on household energy use.

Please see full response to Reviewer 1, Comment #3 as there is significant overlap. Since we have empirical locational data of these households, we are able to perfectly match to the building models on climate zone. In regards to how climate zone is taken into account in the simulated load profiles and the resolution of the model, we have added text to better explain NREL's EULP model. We made edits to include a citation to NREL's technical report outlining their assumptions, calibration, validation, and uncertainty. In addition, we have added language that highlights some main critical points to improve trust in the source. The EULP building models are specific not only to climate zone, but to other geographical specifications such as metropolitan area, county, etc. The building models take into account downscaled weather files specific not only to climate zones, but the locations of these building models (calibrated with empirical meter data with known locations). As such, climate zones (and far more granular in terms of geography) are addressed in these energy use curves. Please see response to R1's Comment #3 for the new text added.

6. Solar profile is at a county level, but what about homes that cannot support solar due to local shading or roof issues? Could they still reduce their energy burden and how does this study demonstrate that?

Our study relies on empirical rooftop solar adoption data. As such, has adopted solar- leaving zero "homes that cannot support solar". Households that cannot adopt rooftop solar can reduce EB through other measures such as energy efficiency or community solar, but those measures are outside the scope of our study.

We understand the reviewer's concern about the variability of solar production across specific locations within each county. To determine the relative importance of PV production variability at specific locations, we have included a stochastic soiling loss factor parameter in our Monte Carlo simulation. Please see the first response to R2's Comment #1 for the added text from new Supplemental Information 8 "Statistics" that shows that our results are robust to different levels of rooftop solar panel soiling and shading.

7. Not enough details in the methodology section are reported to replicate the study as there appear to be flaws when interpreting the statistical results. A decomposition analysis is recommended.

We apologize but this comment is too unclear to be addressed. The reviewer appears to refer to three unrelated issues. The question of whether a methodology is replicable is completely independent of the interpretation of results, and decomposition analysis is a method for understanding drivers within a system. If the reviewer could please clarify the point being made, we could address this if there is another round of review.

8. I doubt the effectiveness of using linear regression to capture the relationship between these potential influencing factors. Have you conducted any statistical tests to prove your choice of

model? Did you compare other models or find some ways to prove the robustness of your regression results?

In Supplemental Information 7, we significantly edited the entire section to reiterate that the results are modeled, such that the influence of the factors is not “potential” but rather ensured in the model design. The goal of the regression was to better understand how these factors work statistically in the model. Conceptually, we could build a complex model that precisely specifies how every variable determines the change in EB. We believe the regression provides a simpler and more intuitive presentation that approximates the deterministic model. To address the reviewer’s comments, we added text to Supplemental Information 7 clarifying that the model is not empirical and is not designed for statistical identification. We specify that the objective of the regression results is only to describe the relative importance of the different factors in the modeled results. We simplified the presentation by specifying the model in fixed rather than random effects. We also added a linear-log specification for comparison to address the comparison to other models mentioned by the reviewer. We find that this does prove the robustness of our regression results. The full edited text is pasted below:

“We wanted to better understand any patterns in adopters’ net energy savings, especially to identify what may contribute to solar costs outweighing revenue for 2021. To do so, we define the dependent variable as the difference between EB with and without solar. A positive value indicates a percentage point reduction in EB due to solar adoption, while a negative value indicates that solar was not economical for the study period. We then implement two specifications of linear regressions to assess statistical associations between changes in EB and other variables:

- (1) An ordinary least-squares regression with standardized inputs (i.e., scaled by subtracting each by the mean and dividing by the standard deviation such that 0 indicates the mean and +/-1 indicates one standard deviation above/below the mean), fuel type, and regional fixed effects*
- (2) A log-linear specification of the same model. Here, the logged variables are more strongly collinear. In order to reduce collinearity, the variable with the least explanatory power in the first model (annual electricity bill) was dropped.*

To reiterate, the dependent variable in these models (change in EB) is a modeled result. These regressions are not empirical models with exogenous errors. Still, the regressions help describe an approximate ranking of the importance of different factors in explaining changes in EB in our model. Greater reductions in EB due to solar were linked to (in order of importance based on standardized coefficients in regressions (1) and (2)): high energy burden pre-solar, high volumetric energy costs, smaller solar sizes, cheaper solar per-Watt prices (after incentives), higher square footage, lower incomes, and larger annual bills (Supplementary Table 10). Although the log-linear coefficients are not directly comparable against each other, the ordering of magnitude of those coefficients is roughly consistent with the ranking just described. Across regions, the models suggest that larger EB reductions were more likely in the Midwest and less likely in the South or West, all else equal. Across heating fuels, households utilizing fuel oil or propane were least likely to see large EB reductions in 2021 due to solar adoption (Supplementary Table 10).

Supplementary Table 10: Regression results (Y=reduction in EB due to solar), standard errors in parentheses

	(1) Standardized Linear	(2) Log-Linear
EB	0.73* (0.027)	1.02* (0.01)
\$/kWh	0.49* (0.005)	1.19* (0.012)
PV Size (kW)	-0.36* (0.004)	-0.79* (0.008)
\$/W	-0.35 (0.003)	-0.9* (0.009)
Square Footage	0.15* (0.004)	0.47* (0.01)
Income	-0.08* (0.013)	0.04* (0.014)
Total Annual Bill	0.05* (0.008)	
Fuel: Oil	-0.81* (0.017)	-0.8* (0.017)
Fuel: Gas	-0.37* (0.014)	-0.32* (0.014)
Fuel: Propane	-0.91* (0.052)	-0.94* (0.048)
Fuel: Other	0.01 (0.129)	0.12 (0.135)
Fuel: None	-0.24* (0.14)	-0.19 (0.135)
Northeast	-0.28* (0.036)	-0.33* (0.04)
South	-0.84* (0.039)	-0.5* (0.044)
West	-0.79* (0.036)	-0.65* (0.04)

*p<0.05

9. What do you mean "in that order of magnitude"? Do you mean the numbers of coefficients? If your independent variables have different units then the coefficients are not comparable. You can use decomposition methods to find the exact contribution of each factor.

*We appreciate Reviewer 2 casting light on an omission on our part. We are able to compare these coefficients because they are scaled in keeping with best practices. We had omitted that detail and it is now added into the body of Supplemental Information 7. It now reads (additions in **bold**): "We then ran a linear regression on scaled independent variables (scaled by subtracting each by the mean and dividing by the standard deviation such that 0 indicates the mean and +/-1 indicates one standard deviation above/below the mean) and found that this level of savings was linked to various variables. With scaled variables, we were able to compare the relative influence of each. "*

10. On page 4 line 199, could you also compare energy burden differences for households with different housing conditions?

Due to data constraints, we are unable to explore energy burden differences across different housing conditions in this study.

11. Energy burden impacts in 2021 based on year of adoption. It seems that you are using 2021 energy prices, have you converted 2013-2020 income to the 2021 price level for comparison to account for inflationary changes?

This may be another misunderstanding. Income estimates are for 2021 to capture the snapshot energy burden impacts of rooftop solar (the income at time of adoption does not factor into 2021 energy burden if adopted during any other year). In the original submitted version, this is explained once each in Results section "Energy burden impacts of rooftop solar in 2021 by income group", the Discussion, Methods, and several times in the Supplemental Information body as well as tables such as Supplemental Table 1. That said, these sections all occur later in the paper and, in order to make this more clear upfront, we have added language in the Abstract, Introduction, and added "2021" before "income estimate" in several locations in the Methods as well as the Results sub-headers in order to reduce any reader confusion upfront.

*For example, two edits in the abstract (additions in **bold**) include "...reduces energy burden across a majority of adopters **during our study period** from a median of..." and later in the abstract: "solar reduced median **2021** energy burden from..."*

*In the Introduction, we have added the following to the last paragraph: " In this study, we consider a broader scope of financial impacts when estimating EB **during our study period of 2021**"*

We hope that these edits, especially those upfront in the Abstract, Introduction, and Results sub-headers make more clear that this study is leveraging empirical 2021 data and 2021-specific household-level estimates to provide results for a specific study period.

12. You mention regional differences, but keep referring to SI 7. I suggest conducting regressions by regions to prove "regional difference" instead of concluding your results from overall regression. This is misapplying the downscaling of the statistical results.

This comment is related to Comment #8. To clarify once more, these are not empirical results, we are not attempting here to "prove" regional results, they are an observed output of the modeled results. The regressions provide just one way of visualizing those differences. If we define "regional results" as differences in the change in EB across regions, then the regression presented in our original Supplemental Information 7 provides evidence of those regional differences. The reviewer seems to imply a different type of regional difference: differences in the influences of the different variables across regions, which is what would be measured by conducting separate regressions across regions. While regional interactions may be of interest in an empirical study, we believe they are out of scope in our study.

13. You should also report variable statistics such as mean, sd, min, and max instead of only reporting variable types. (Table 1)

Table 1 has been updated with these recommendations for numeric variables.

Reviewer 3

1. Providing supporting references in the first paragraph on page 2 would bolster the credibility of the argument that the burden of air pollution and cost shifts is uneven.

We have added citations to address this comment. For the air pollution burden, we cite Sovacool and Dworkin (2014). For the cost shift, we now cite Borenstein (2017). Both are highly-cited and foundational texts on these two topics.

2. The examples provided (low-income households using fuel oil, African American households in mixed-humid climate zones, etc.) seem to lack a direct connection with health impacts and are more related to socioeconomic, housing, and built environment characteristics. Suggest removing the term "health impacts" from the sentence as the health impacts were introduced in the subsequent sentences.

We have applied this suggestion to the second paragraph in "Introduction" as well as simplified the section for clarity- removing any redundant or confusing language.

3. While it is commendable to see the limitation of the definition "energy burden" and the rationale for focusing on this topic, I recommend moving this information to the study's limitations section, as it aligns more with future research opportunities.

We implement this suggestion, and this has been moved to Discussion, paragraph 5.

4. Providing supporting references for the argument that existing literature on energy burden reduction from solar focuses solely on bill impacts, without accounting for ongoing off-bill financial impacts of solar adoption, would strengthen the narrative. If available, comparing these references with the study's results could emphasize the significance of the current study.

We have cited Cook and Shah for this statement. We also strengthen our statement that this paper fills additional gaps in the literature by including an additional reference that came out since initial submission (Hardy et al. 2024) to show that other studies focus on only a few locations and rely heavily (solely) on aggregated and/or modeled data.

We hope that this addresses not only this comment, but other reviewer comments (both Reviewer 1 and 2) on our use of modeled end-use load curves by highlighting that we use empirical data wherever possible (energy costs, property information, solar installation data) and, where modeled data are necessary, we use address-specific estimates where available (e.g., income) and other data as granular as possible, calibrated with empirical data (e.g., property and housing characteristics). In addition, our study spans multiple states in the U.S. and encompasses 0.5M households, providing a much larger sample than other studies.

5. The assumption that all solar adoption credit benefits go to the third party, leading to reduced prices for customers in the form of lower lease rates, may not always hold true. A sensitivity analysis, considering the magnitude of credit pass-through, would be valuable for comparing impacts between lease and loan options. This is crucial as the results for the lease option could potentially be underestimated.

This is related to Reviewer 2's Comment #4. We ran separate simulations allowing only the lease and loan assumptions to vary over 1,000 iterations for each. These simulations allow us to isolate the sensitivity of the impacts on EB to underlying assumptions around leasing and loans. The results of those simulations are depicted in Supplemental Figure 4. That figure suggests that the results are generally more sensitive to our leasing assumptions, while varying loan assumptions within reasonable bounds has relatively little impact on the results. In the main text, we reference this finding and where to go in the Supplemental Information in the second paragraph of Results Section "Energy burden impacts of rooftop solar in 2021 based on ownership model". Please see response to R2's Comment #4 for reproduction of new text, table, and figure included to address these suggestions.

6. It would be beneficial to explain the decreasing pattern of energy burden with solar from 2020 to 2021 in Figure 5, as this trend is quite apparent. Providing context for this observation would enhance the reader's understanding.

Since this is one year of change (note that year is "year of adoption" as labeled on the axis and all EB reduction pertains to 2021), it is hard to tell if this is a true trend. As such, we are unsure if this will persist to a true "trend" and we fear that any explanation may be too speculative or reductive (see dip in 2016 adopters). As such, we do not address this comment.

REVIEWER COMMENTS

Reviewer #1 (Remarks to the Author):

The authors have provided evidence to support some of the constructive feedback provided, and I thank them for that.

Their analysis is precise regarding who installs solar, where, how much they earn, and building characteristics. Still, due to limitations, the authors broadened the linkage with how much electricity they consume –a fundamental piece to answer their research question precisely– and relied on modeled data from EULP / ResStock.

At the very least, the title of the article should be reconsidered to state “Modeling the potential effects of rooftop solar on household energy burden in the United States” rather than a conclusive title that would be best suited for an analysis that has consumer-level information provided by the utilities, as stated as a limitation in the article and throughout the responses to Reviewers 2 and 3.

Reviewer #2 (Remarks to the Author):

The authors have done a commendable job considering the reviewer comments from the first round of reviews and I believe have addressed the previous concerns.

We appreciate all three reviewers and are happy that we were able to address all of R2's concerns and the majority of R1's concerns. Below please find our response to the one remaining comment, which we hope will address the remaining concerns of R1.

Reviewer 1

1. Their analysis is precise regarding who installs solar, where, how much they earn, and building characteristics. Still, due to limitations, the authors broadened the linkage with how much electricity they consume –a fundamental piece to answer their research question precisely– and relied on modeled data from EULP / ResStock. At the very least, the title of the article should be reconsidered to state “Modeling the potential effects of rooftop solar on household energy burden in the United States” rather than a conclusive title that would be best suited for an analysis that has consumer-level information provided by the utilities, as stated as a limitation in the article and throughout the responses to Reviewers 2 and 3.

We have taken this suggestion directly and have changed the title as such. We do not intend to be misleading about our methods and believe that this change addresses R1's concern.